# Transactional sex among women in Sub-Saharan Africa: A systematic review and meta-analysis

**Gedefaye Nibret Mihretie**[1]*, **Bekalu Getnet Kassa**[1], **Alemu Degu Ayele**[1], **Tewachew Muche Liyeh**[2], **Habtamu Gebrehana Belay**[1], **Agernesh Dereje Miskr**[1], **Binyam Minuye**[2], **Melkalem Mamuye Azanaw**[3], **Mulugeta Dile Worke**[1]

**1** Department of Midwifery, College of Health Sciences, Debre Tabor University, Debre Tabor, Ethiopia, **2** School of Public health at University of Technology Sydney, Sydney, Australia, **3** Department of Social and Public Health, College of Health Sciences, Debre Tabor University, Debre Tabor, Ethiopia

* gedefayen@gmail.com

## Abstract

### Introduction

Transactional sex is casual sex between two people to receive material incentives in exchange for sexual favors. Transactional sex is associated with negative consequences, which increase the risk of sexually transmitted diseases, including HIV/AIDS, unintended pregnancy, unsafe abortion, and physiological trauma. In Sub-Saharan Africa, several primary studies have been conducted in various countries to examine the prevalence and associated factors of transactional sex among women. These studies had great discrepancies and inconsistent results. Hence, this systematic review and meta-analysis aimed to synthesize the pooled prevalence of the practice of transactional sex among women and its associated factors in Sub-Saharan Africa.

### Method

Data source: PubMed, Google Scholar, HINARI, the Cochrane Library, and grey literature were searched from March 6 to April 24, 2022, and included studies conducted from 2000 to 2022. The pooled prevalence of transactional sex and associated factors was estimated using Random Effect Model. Stata (version 16.0) was used to analyze the data. The I-squared statistic, a funnel plot, and Egger's test were used to check for heterogeneity and publication bias, respectively. A subgroup analysis was done based on the study years, source of data, sample sizes, and geographical location.

### Results

The pooled prevalence of transactional sex among women in Sub-Saharan Africa was 12.55% (9.59%–15.52%). Early sexual debut (OR = 2.58, 95% CI: 1.56, 4.27), substance abuse (OR = 4.62, 95% CI: 2.62, 8.08), history of sexual experience (OR = 4.87, 95% CI: 2.37, 10.02), physical violence abuse (OR = 6.70, 95% CI: 3.32, 13.53), orphanhood (OR = 2.10, 95% CI: 1.27, 3.47), and sexual violence abuse (OR = 3.76, 95% CI: 1.08, 13.05) were significantly associated with transactional sex.

**Data Availability Statement:** All available data are found in the paper.

**Funding:** The authors received no specific funding for this work.

**Competing interests:** The authors have declared that no competing interests exist.

**Abbreviations:** DHS, Demography and Health Survey; TS, Transactional Sex.

## Conclusion

The prevalence of transactional sex among women in sub-Saharan Africa was high. Alcohol consumption, substance abuse, early sex debuts, having a history of sexual experiences, physical violence, and sexual violence increased the practice of transactional sex.

## Introduction

Transactional sex is defined as a sexual act (s) that is structured by the implicit assumption that sex is exchanged for a variety of instrumental supports such as educational expenses, transportation, a place to sleep, clothing, material items, or money. It reveals that socioeconomic factors have a great role in establishing exchange-based sexual relationships in many countries with high HIV prevalence. Commonly, in many countries, men provide and women receive material rewards [1, 2].

Globally, an estimated 36.7 million individuals worldwide were infected with the Hunan Immunodeficiency Virus (HIV) epidemic. Young people in sub-Saharan Africa (SSA) are more threatened by the HIV epidemic than young people elsewhere. In total, there were 2.1 million new HIV infections worldwide, with 1.1 million occurring in sub-Saharan Africa [3]. HIV infection deaths mainly affect the young and productive segments of the community. Among newly infected people in SSA, 40% belong to the age group of 15–24 years, and more than 60% of these infections occurred among young girls [4].

Significant proportions of females have multiple concurrent sexual relationships and engage in risky sex. Because of their risky sexual *practices*, the girls and their sexual partners, including schoolmates, are at risk of HIV infection and other sexually transmitted infections (STIs) [5, 6]. Among HIV-infected young people in the world, 63% lived in sub-Saharan Africa, and among these, 59% were female. Unprotected sex also puts women at risk of unintended pregnancy, which leads to unsafe abortions [7, 8]. Sex motivated by financial gain is a serious public health issue, particularly in sub-Saharan Africa [9].

Assessing transactional relationships is still an important aspect of HIV prevention initiatives [10]. Individual behaviours that harm people's chances of acquiring sexually transmitted diseases (STDs) and unwanted pregnancies were identified. Among these identified risk behaviours, transactional sex (sex in exchange for money, gifts, benefits, or other monetary rewards) is the main one [11]. The infection rate of STIs, including HIV, among young women aged 15 to 24 years old, is greater than that of young males (3.6 to 1 ratio) [12]. Early sexual activity, early pregnancy, unsafe abortions, and the increase in HIV infections have become major concerns in sub-Saharan Africa [13]. Unwanted pregnancy is the cause of school dropout in girls. School dropout is an additional barrier that women, severely handicapped by parenthood, must face to overcome the longer-term impacts of childbearing [14]. Transactional sex involves engaging in sex for money or gifts in order to increase one's long-term life chances [15, 16].

In sub-Saharan Africa, cultural and social norms, gender inequality, and harmful traditional *practices*, combined with a lack of access to reproductive health services, a high unemployment rate, and young females from lower-income families, expose young people to a variety of social and economic challenges and encourage them to engage in transactional sex [17, 18].

Transactional sex occurred at a rate of 2% in Niger, 14% in Benin, 14% in Kenya, 27% in Zambia, 31% in Uganda, 5% in Cameroon, and 85–90% in Uganda among sexually active girls who reported ever engaging in sexual relations in exchange for money or gifts in the last 12 months [15, 19–23]. Transactional sex is associated with HIV risk factors or behaviours

including alcohol use [24], sexual or physical violence or abuse [25], inconsistent condom use [26] and multiple partners [27].

In Sub-Saharan Africa, several primary studies have been conducted in various countries to examine the prevalence and associated factors of transactional sex among women. These studies had great discrepancies and inconsistent results across countries. Hence, this systematic review and meta-analysis aimed to synthesise the pooled prevalence of the *practice* of transactional sex among women and its associated factors in sub-Saharan Africa.

## Methods

### Study design and settings

This meta-analysis and systematic review were carried out in Sub-Saharan African countries (Angola, Benin, Botswana, Burkina Faso, Burundi, Cameroon Cape Verde, Chad, Central African Republic, Comoros, Congo, Côte d'Ivoire, Djibouti, Equatorial Guinea, Eritrea, Ethiopia, Gabon, Gambia, Ghana, Guinea, Guinea-Bissau, Kenya, Lesotho, Liberia, Madagascar, Malawi, Mali, Mauritania, Mozambique, Namibia, Niger, Nigeria, Rwanda, Senegal, Seychelles, Sierra Leone, Somalia, South Africa, Sudan, Tanzania, Togo, Uganda, Western Sahara, Zambia, Zimbabwe). The International Prospective Register of Systematic Reviews has included this review in the protocol (CRD42022323168).

### Data source and search strategy

This review and meta-analysis were developed based on the PRISMA (Preferred Reporting Items for Systematic Review and Meta-Analysis) guidelines [28]. Studies on transactional were identified through an online search of PubMed, HINARI, Google Scholar, the Cochrane Library, and grey literature. Articles were searched from March 6, 2022, to April 24, 2022 (**S1 File**).

**Eligibility criteria ("PPECOLD"). Population (P):** The study participants were women aged between 10 and 55 in sub-Saharan Africa. The included studies were from all socioeconomic statuses, all ethnic groups, and all languages but reported in the English language. Women who had casual sex with men in exchange for money, materials, or any benefits in exchange for sexual favours within the past 12 months.

**Publication year (P):** We planned to investigate the cumulative prevalence of transactional sex and its determinants from the beginning of the Millennium Development Goals in 2000 until the end of our data search in 2022 (January 1, 2000, to March 28, 2022).

**Exposure (E):** factors associated with transactional sex and at least two times reported as significant factors (socio-demographics such as age and educational status), participants' having not either one or both parents, alcohol use, substance abuse, an early sex debut, history of sexual experiences, physical violence, and sexual violence (**Table 1**).

**Comparison (C):** The reported reference groups for each determinant factor in each respective study, such as substance abuse participants versus those who did not abuse.

**Outcome measurement (O):** The magnitude and associated factors of transactional sex.

**Language (L):** All included studies were reported in the English language.

**Design (D):** case-control and cross-sectional studies were assessed.

**Exclusion criteria:** citations without full texts, duplicate studies, anonymous reports, case reports, and qualitative studies were excluded.

### Screening and data extraction

This study includes both published and unpublished articles on the magnitude of transactional sex and associated factors among women in Sub-Saharan Africa. All search articles were

**Table 1. Population exposure comparison and outcome variable (PECO) summary table.**

| Population | Exposure | Comparison | | Outcomes |
|---|---|---|---|---|
| Women | Orphanhood | Women have neither one nor both parents | Women have both parents* | Transactional sex among women |
| Women | Age | Women age ≥18 years | Women age < 18 years* | Transactional sex among women |
| Women | Educational status | women have formal education | Participants have no formal education* | Transactional sex among women |
| Women | Alcohol use | Alcohol user | Non-alcohol user* | Transactional sex among women |
| Women | Substance abuse | Substance abuser(chat chewing, cocaine, cigarette smoking, morphine, shisha) women | Non-users* | Transactional sex among women |
| Women | Early sex debut | Women aged less than 16 years | Woman's age greater than or equal to 17 years* | Transactional sex among women |
| Women | Having a history of sexual experiences | Women who had a sexual history before they engaged in transactional sex | Women who had no sexual history before they engaged in transactional sex* | Transactional sex among women |
| Women | Physical violence | Women who had a history of physical violence before engaging in transactional sex | Women who did not have a history of physical violence before engaging in transactional sex* | Transactional sex among women |
| Women | Sexual violence | Women who had a history of sexual violence before engaging in transactional sex | Women who did not have a history of sexual violence before engaging in transactional sex* | Transactional sex among women |

* = Reference Group

exported to the Endnote X7 reference manager software, and duplicated articles were excluded. The articles were screened and assessed after carefully reading the titles and abstracts by nine authors (GNM, BGK, ADA, TML, HGB, ADM, BM, MMA, and MDA) independently. The full text of the studies was further evaluated based on objectives, methods, population, and outcomes. Disagreements between authors were resolved through discussion and consensus based on quality assessment tool.

Following the selection of eligible studies, the authors independently extracted all necessary data using a standardized data extraction form. This form includes the primary author, study year, year of publication, study setting, sample size, study design, prevalence, and each specific factor associated with transactional sex. Selected variables had at least two or more studies reporting them as significant factors.

**Quality assessment.** The scientific strength and quality of each study were assessed by using the Newcastle-Ottawa Scale quality assessment tool [29]. All authors independently, using the assessment tool, weighted the qualities of each original study. An assessment that scores 50% or above was included for analysis (≥5 out of 10). Score differences between the investigators were managed by taking the average score of their quality evaluation outcomes (**S2 File**).

**Publication bias and heterogeneity.** Comprehensive searches (database and manual searches) were used to minimise the risk of bias. The authors' cooperative work was also crucial in reducing bias, selecting articles based on clear objectives and eligibility criteria. A visual inspection of the funnel plot graph and Egger's tests at a 5% significant level were done to assess the presence of publication bias [30, 31]. Point estimation and subgroup analysis were used to analyse the random variations among the primary studies. I-squared statistics with corresponding p-values were used to assess heterogeneity across and within studies.

**Statistical analysis and data presentation.** We used Microsoft Excel for data entry and STATA-16 software for analysis. The random-effects model (DerSimonian-Laird method) was considered to assess for variations between the studies. the data was summarised by pooled prevalence and odd ratio. The results were presented using texts, tables, and forest plots with measures of effect and a 95% confidence interval.

## Results

### Study selection

Four thousand one hundred seventy-five primary studies were identified by using the major medical and health electronic databases and registers. The seven studies were from other relevant sources. From the 4182 identified studies, 1124 were excluded after reviewing their titles due to duplication, whereas 2827 articles were allowed further screening. Of the remaining 231 articles, 199 were excluded due to a non-targeted population, an inconsistent study report, the outcome of interest not being reported, the unavailability of full text, and inconsistency with the predetermined inclusion criteria for the review. Finally, 32 studies were used for the systematic review and meta-analysis, with a total population of 108,075 (**Fig 1**).

### Characteristics of the included studies

All 32 eligible studies were reported in English. The sample size ranges from 204 in Nigeria [32] to 8984 in Malawi [33]. Based on the geographical location, three studies were from Ethiopia [10, 34, 35], three studies were from Uganda [36–38], one was from Liberia [39], three were from South Africa [40–42], four were from Nigeria [32, 43–45], one was from Cameron [46], two were from Malawi [33, 44], two studies were from Kenya [44, 47] and one study was from each country (Zambia, Zimbabwe, Benin, Burkina Faso, Central Africa Republic(CAR), Chad, Guinea, Niger, and Togo) [44]. The included studies dealt with practises of transactional sex among women and associated factors in sub-Saharan African Countries [10, 32–47] (**Table 2**).

### The magnitude of transactional sex

The pooled prevalence of transactional sex among women in Sub-Saharan African countries was 12.55% (95%CI: 9.59%, 15.52%) (**Fig 2**).

### Heterogeneity and publication bias

This study had heterogeneity ($I^2$ = 99.60%, $P \leq 0.001$). Publication biases were examined by using both funnel plots and Egger's regression test. The results of funnel plots showed an asymmetric shape, which indicates the presence of publication bias (**Fig 3A**). Egger's regression test also showed the presence of publication bias across studies (p-value <0.001). The nonparametric trim and fill analyses were done after examining the publication bias. Trimming and filling analysis was used to fill in 16 missing studies in the funnel plot to correct the publication bias. After imputed 16 studies from 32 observed studies, the pooled prevalence was 4.93% (95% CI: 1.82%-8.03%) using the random effect model (**Fig 3B**).

### Sensitivity analysis

To determine the potential source of heterogeneity seen among the eligible studies, the authors did a sensitivity analysis. The sensitivity analysis result indicated that the source of heterogeneity did not depend on a particular study (**Fig 4**).

### Subgroup analysis

Subgroup analysis was done based on publication years, the number of sample sizes, the source of the data, and the geographical location. Based on publication year, the lowest prevalence was from 2000 to 2005 years (4.34%, 95% CI: 3.16%, 5.51%), and the highest prevalence was from 2011 to 2015 years (32.77%, 95% CI: 5.00%, 60.54%) (**Fig 5**). The studies with less than

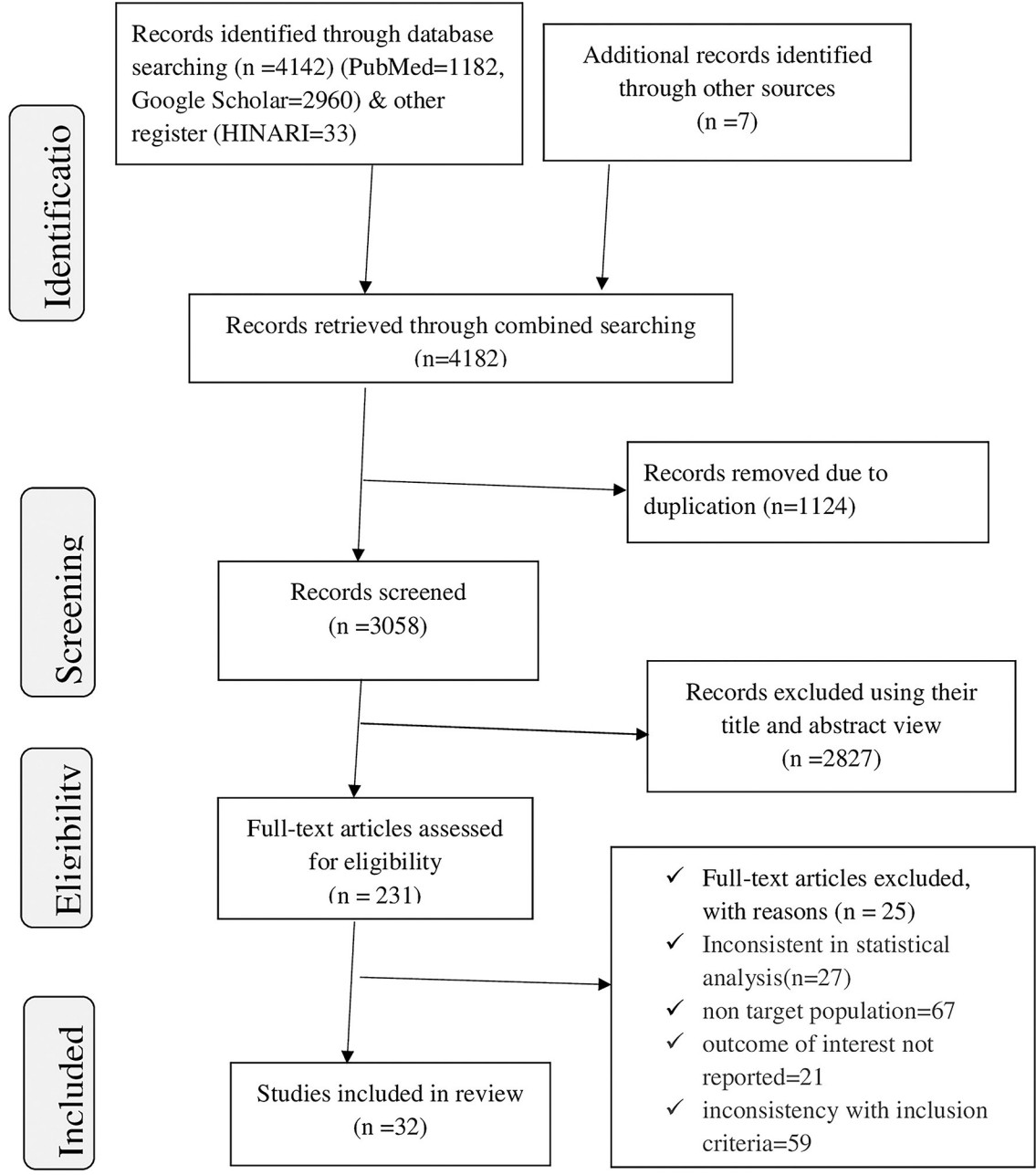

**Fig 1. PRISMA flow chart revealing study selection for systematic review and meta-analysis of prevalence and associated factors of transactional sex among women in Sub-Saharan Africa.**

3000 sample sizes have the highest pooled prevalence of transactional sex (18.41%, 95% CI: 12.82%–24.00%) (**Fig 6**). The highest pooled prevalence of transactional sex was found in studies conducted in a single study area (21.02%, 95% CI: 9.59%–15.52%), and the lowest pooled prevalence was found in studies using Demography and Health Survey (DHS) data (4.34%, 95% CI: 3.16%–5.51%) (**Fig 7**). According to the geographical region classification, East Africa had the highest prevalence (15.57%, 95% CI: 11.38%, 19.75%) (**Fig 8**).

**Table 2. Characteristics of included studies reporting the magnitude and associated factors of transactional sex among women in Sub-Saharan Africa, 2000 to 2022.**

| First Author & Year of publication | Country | Study design | Woman's age | Reference period | Sample size | Prevalence | Outcome measurement |
|---|---|---|---|---|---|---|---|
| Kassa AW et al. (2018) | Ethiopia | Cross-sectional | 17–19 old years | TS within the last 12 months | 726 | 17.6% | Magnitude & associated factors of transactional sex |
| Dana LM et al. (2019) | Ethiopia | Cross-sectional | 15–24 year old | TS within the last 12 months | 634 | 11.5% | Transactional sex and HIV risks |
| Stamatakis C. et al. (2021) | Uganda | National survey | 13–24 years old | TS within the last 12 months | 1515 | 14.2% | Regional heterogeneity and associated with transactional sex |
| Okigbo CC et al. (2014) | Liberia | Cross-sectional | 13–24 years old | TS within the last 12 months | 493 | 72.0% | Magnitude and risk factors of transactional sex |
| Duby Z et al. (2021) | South Africa | Cross-sectional | 13–24 years old | TS within the last 12 months | 4399 | 12.1% | Motivations for Engaging in Transactional Sex |
| Ajayi AI et al. (2019) | Nigeria | cross-sectional | 14–25 years old | TS within the last 12 months | 630 | 17.9% | The magnitude and associated factors of transactional sex |
| Ranganathan M et al. (2016) | South Africa | Cross-sectional | 15–24 year old | TS within the last 12 months | 693 | 14.0% | Transactional sex prevalence, mediators, and association with HIV infection |
| Akoku DA et al. (2018) | Cameron | Cross-sectional | 21-49-year-olds | TS within the last 12 months | 506 | 14.9% | Socio-economic vulnerabilities and HIV: Drivers of transactional sex |
| Choudhry V et al. (2014) | Uganda | cross-sectional | 13–20 year old | TS within the last 12 months | 867 | 25.0% | Giving or Receiving Something for Sex |
| Gichane MW et al. (2022) | Malawi | Cross-sectional | ≥21 years old females | TS within the last 12 months | 920 | 22.0% | Individual and Relationship-Level Correlates of Transactional Sex |
| Animasahun VJ et al. (2019) | Nigeria | cross-sectional | 15-49-year-olds | TS within the last 12 months | 204 | 7.4% | Transactional Sex among Women Accessing Antiretroviral Treatment |
| Chatterji M et al. (2005) | Kenya | DHS data extraction | 15–49 years old | TS within the last 12 months | 6612 | 6.7% | The Factors Influencing Transactional Sex in 12 Sub-Saharan African Countries |
| Chatterji M et al. (2005) | Zambia | DHS data extraction | 15–49 years old | TS within the last 12 months | 7128 | 11.0% | The Factors Influencing Transactional Sex in 12 Sub-Saharan African Countries |
| Chatterji M et al. (2005) | Zimbabwe | DHS data extraction | 15–49 years old | TS within the last 4 weeks | 4920 | 3.6% | The Factors Influencing Transactional Sex in 12 Sub-Saharan African Countries |
| Chatterji M et al. (2005) | Benin | DHS data extraction | 15–49 years old | TS within the last 12 months | 4951 | 3.7% | The Factors Influencing Transactional Sex in 12 Sub-Saharan African Countries |
| Chatterji M et al. (2005) | Burkinafaso | DHS data extraction | 15–49 years old | TS within the last 12 months | 5610 | 1.8% | The Factors Influencing Transactional Sex in 12 Sub-Saharan African Countries |
| Chatterji M et al. (2005) | CAR | DHS data extraction | 15–49 years old | TS Within the last4 weeks | 5342 | 3.8% | The Factors Influencing Transactional Sex in 12 Sub-Saharan African Countries |
| Chatterji M et al. (2005) | Chad | DHS data extraction | 15–49 years old | TS within the last 12 months | 6593 | 2.5% | The Factors Influencing Transactional Sex in 12 Sub-Saharan African Countries |
| Chatterji M et al. (2005) | Guinea | DHS data extraction | 15–49 years old | TS within the last 12 months | 6135 | 3.7% | The Factors Influencing Transactional Sex in 12 Sub-Saharan African Countries |
| Chatterji M et al. (2005) | Malawi | DHS data extraction | 15–49 years old | TS within the last 12 months | 8984 | 6.5% | The Factors Influencing Transactional Sex in 12 Sub-Saharan African Countries |
| Chatterji M et al. (2005) | Niger | DHS data extraction | 15–49 years old | TS within the last 12 months | 6621 | 1.6% | The Factors Influencing Transactional Sex in 12 Sub-Saharan African Countries |
| Chatterji M et al. (2005) | Nigeria | DHS data extraction | 15–49 years old | TS within the last 12 months | 6871 | 5.5% | The Factors Influencing Transactional Sex in 12 Sub-Saharan African Countries |
| Chatterji M et al. (2005) | Togo | DHS data extraction | 15–49 years old | TS within the last 12 months | 7787 | 2.4% | The Factors Influencing Transactional Sex in 12 Sub-Saharan African Countries |
| Biddlecom AE et al. (2007) | Malawi | National surveys | 15–49 years old | TS within the last 12 months | 1830 | 7.5% | Prevalence and meanings of exchange of money or gifts for sex in sub-Saharan Africa |
| Biddlecom AE et al. (2007) | Burkina Faso | National surveys | 15–49 years old | TS within the last 12 months | 2547 | 11.2% | Prevalence and meanings of exchange of money or gifts for sex in 4 sub-Saharan Africa |
| Biddlecom AE et al. (2007) | Ghana | National surveys | 12–19 years old | TS within the last 12 months | 2111 | 7.2% | Prevalence and meanings of exchange of money or gifts for sex in 4 sub-Saharan Africa |

*(Continued)*

**Table 2.** (Continued)

| First Author & Year of publication | Country | Study design | Woman's age | Reference period | Sample size | Prevalence | Outcome measurement |
|---|---|---|---|---|---|---|---|
| Biddlecom AE et al. (2007) | Uganda | National surveys | 12–19 years old | TS within the last 12 months | 2354 | 9.2% | Prevalence and meanings of exchange of money or gifts for sex in 4 sub-Saharan Africa |
| Alamirew Z et al. (2013) | Ethiopia | Crosse-sectional | 12–19 years old | TS within the last 12 months | 790 | 27.8% | Prevalence and correlates of exchanging sex for money (gift) |
| Chiang L et al.(2021) | Uganda | Crosse-sectional | 12–19 years old | TS within the last 12 months | 1795 | 14.8% | Sexual risk behaviors, mental health outcomes and associated with childhood transactional sex |
| Ige OS et al. (2021) | Nigeria | cross-sectional | 15-49-year-olds | TS within the last 12 months | 239 | 23.85% | Drivers of transactional sexual relationships |
| Becker ML et al. (2018) | Kenya | Crosse-sectional | 18–24 years old | TS within the last 12 months | 1299 | 13.6% | HIV Prevalence, Young Women Engaged in Sex Work, Transactional Sex, and Casual Sex |
| Magni S et al. (2015) | South Africa | National survey | 16-55-year-olds | TS within the last 12 months | 5969 | 6.3% | Alcohol Use and Transactional Sex |

TS = transactional sex

## Factors associated with transactional sex

Nine associated variables were extracted from the primary articles. However, only eight variables were associated with transactional sex. As for educational status, participants' having neither one nor both parents, alcohol use, substance abuse, an early sex debut, a history of sexual experiences, physical violence, and sexual violence were significantly associated with transaction sex. Age of participants greater than 18 years (OR = 1.71, 95% CI, 0.52, 5.62) was not associated with transactional sex by meta-analysis (**Fig 9**).

Education status was significantly associated with transactional sex [10, 39]. Participants who completed primary school and above (OR = 0.48, 95% CI, 0.27, 0.691) were inversely associated with transactional sex as compared to women who did not complete primary school. The heterogeneity test indicated $I^2 = 0.00\%$, $P = 0.92$. Four studies showed that participants who drank/used alcohol were a significant predictor of TS [40, 42, 43, 46].

Study participants who had used alcohol 2.04 times (OR = 2.04, 95% CI, 1.36, 3.05) more likely to *practice* transactional sex than women who did not use alcohol. The heterogeneity test showed that $I^2$ value of 64.41%, $P = 0.04$. Orphanhood (participants' have neither one nor both parents) [34, 36] made twice more likely have transactional sex as compared to women who had both parents (OR = 2.10 95% CI, 1.27, 3.47). The heterogeneity test revealed that $I^2$ value of 0.00%, $P = 0.40$ (**Fig 9**).

Early sexual debut was significantly associated with TS [36, 39]. Participants who had had their first sexual intercourse before or at the age of 16 years were 2.58 times (OR = 2.58, 95% CI, 1.56, 4.27) more likely to have TS as compared to participants who had first sexual intercourse after 16 or later years. The heterogeneity test indicated $I^2 = 0.00\%$, $P = 0.88$. Participants who had a history of sexual experience two and more years before the engagement of TS were 4.87 times more likely to practise transactional sex than women who had no history of sex before the engagement of TS (OR = 4.87, 95% CI, 2.37, 10.02) [39, 46]. The heterogeneity test indicated that $I^2$ value of 54.26%, $P = 0.14$ (**Fig 9**).

Women who had used substances (chat chawing, cocaine, heroin, morphine) were associated with TS [10, 34, 35, 43]. Study participants who had used substances were 4.62 times (OR = 4.62, 95%CI, 2.64, 8.08) more likely to have transactional sex than women who did not use substances. The heterogeneity test showed an $I^2$ value of 64.31%, $P = 0.04$. Women who had a history of physical violence were 6 times (OR = 6.70, 95% CI, 3.22, 13.53) [36, 37] more liked to practice transactional sex than women who did not have physical violence. The

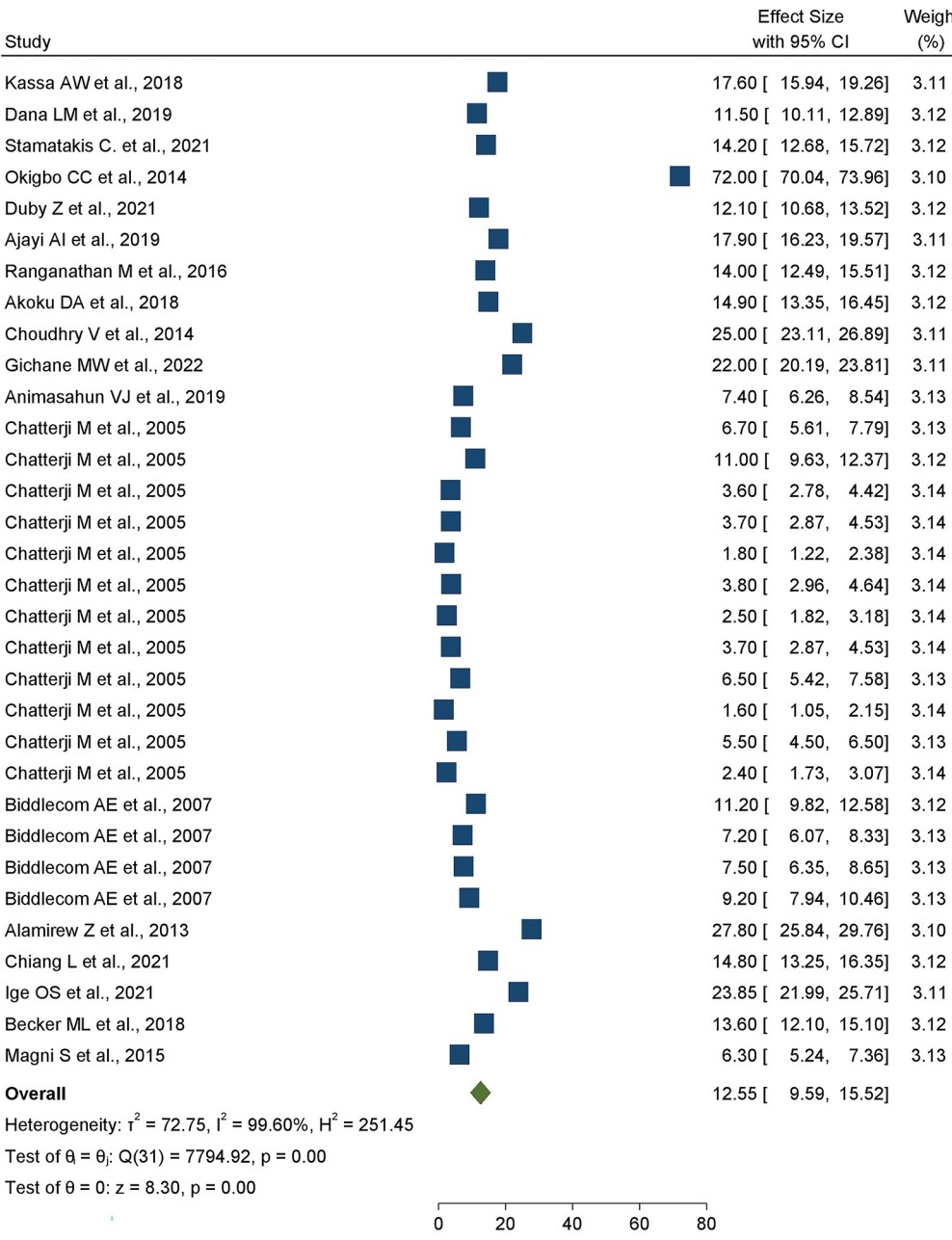

**Fig 2. Forest plot of the pooled prevalence of practices of transactional sex among women in Sub-Saharan countries.**

heterogeneity test showed an $I^2$ value of 0.00%, *P = 0.83*. Sexual violence were a determinant factor for transactional sex (OR = 3.76, 95% CI, 1.08, 13.05) [36, 39]. The heterogeneity test was $I^2$, 64.31%, and p = 0.04 [36, 39] (**Fig 9**).

## Discussion

This systematic review and meta-analysis aimed to synthesize the pooled prevalence of transactional sex and its associated factors among women in sub-Saharan Africa. Thirty-two studies

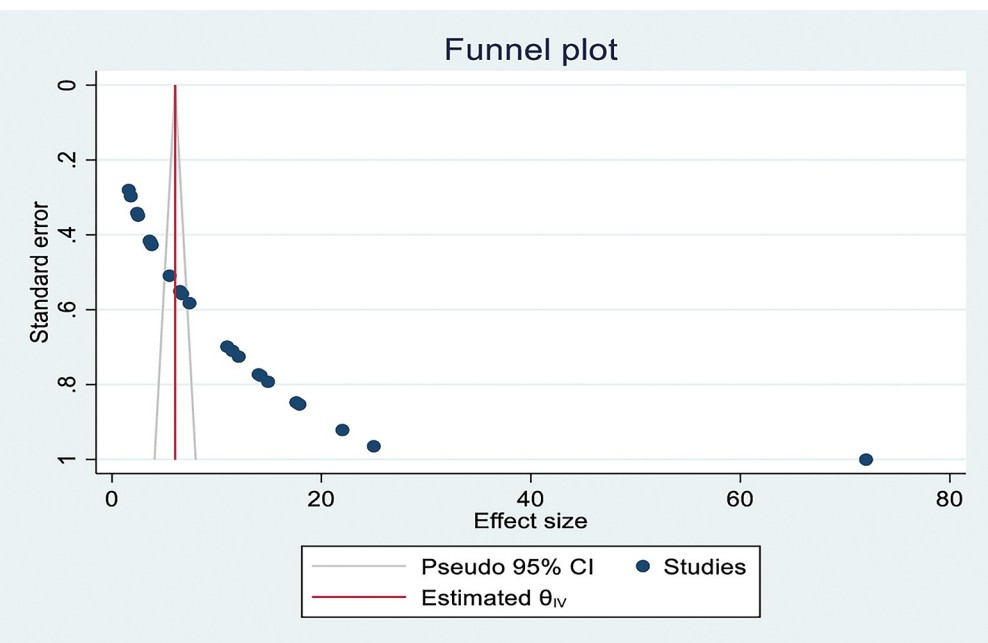

a: Funnel plot for assessing publication bias of the prevalence of transactional sex among women in Sub-Saharan Countries.

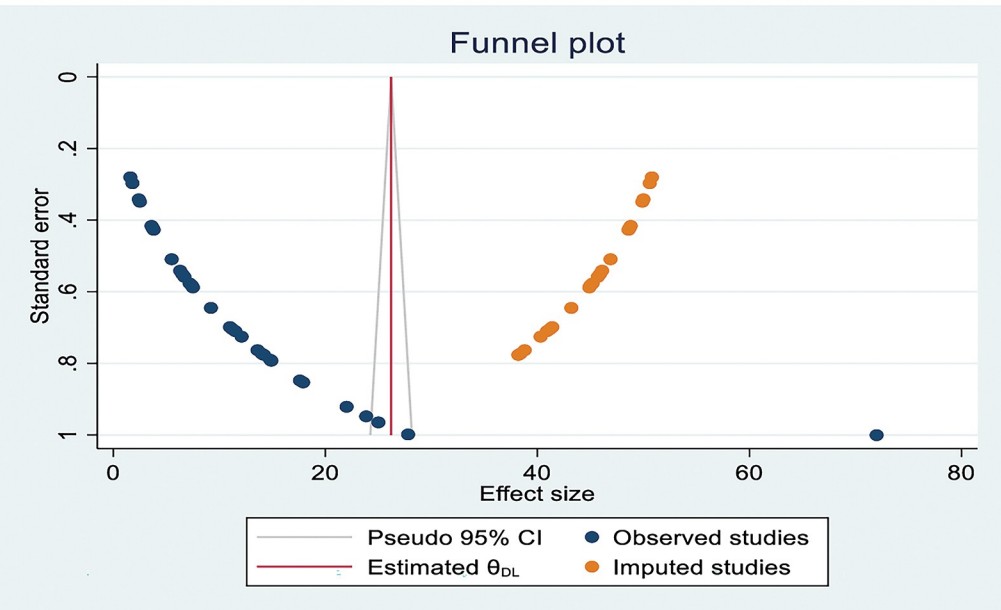

b: Result of trim and fill analysis for adjusting publication bias of the 48 studies

**Fig 3.** a: Funnel plot for assessing publication bias of the prevalence of transactional sex among women in Sub-Saharan Countries. b: Result of trim and fill analysis for adjusting publication bias of the 48 studies.

with 108,075 study participants were included and analysed in this review and meta-analysis. Included studies were conducted between January 1, 2000, and March 28, 2022. The pooled prevalence of transactional sex among women in Sub-Saharan Africa was 12.55% (95% CI: 9.59%, 15.52%). Our findings are comparable to those from a study by Krisch, M., et al. in high-income countries [48]. This finding is also comparable to the study conducted by Dunkle, K.L., et al. on African American women [49]. However, this finding was lower than the

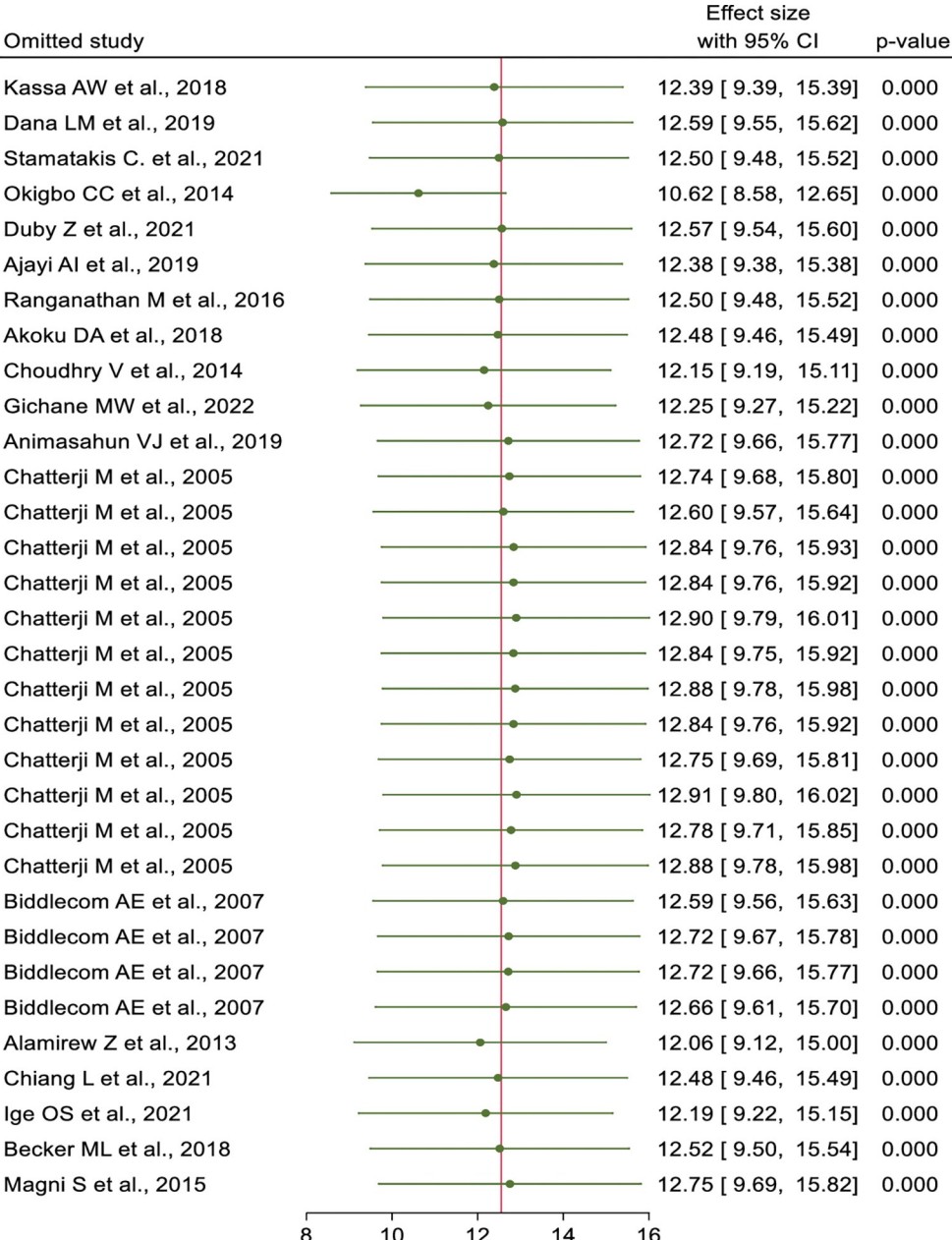

**Fig 4. Sensitivity analyses for the prevalence of transactional sex and associated factors among women in Sub-Saharan Africa.**

following primary studies conducted before 2000 in Sub-Saharan Africa (Cameroon [50], Malawi [51], and Tanzania [52]).

Women in Nigeria, 18% [51]; in Kenya, 78% [53]; in Canada, 7% [54]; In Sweden, 1.5% [55]; in America, 57% [56], and in Norway, 1.4% [57] have ever exchanged sex for money, gifts, or favours. This variation might be due to the difference in the study period, sociodemographic characteristics, socio-economic development variation, geographical area, the definition of transactional sex, and the source of the studied data. In addition, the comparative studies were primary research.

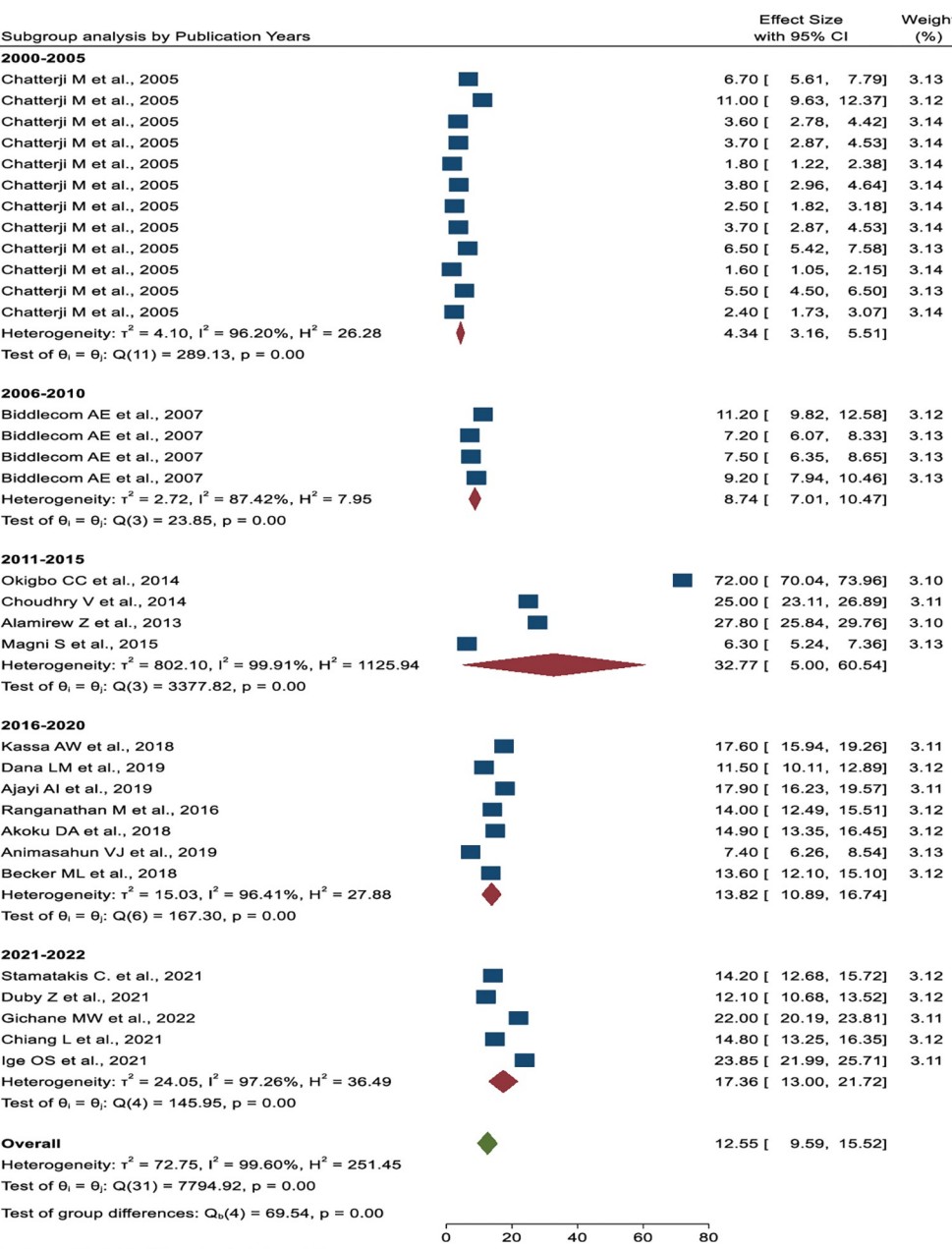

**Fig 5. Subgroup analysis of the pooled prevalence of transactional sex among women based on the study period in Sub- Saharan Africa.**

Studies conducted from 2000 to 2005, observed the lowest pooled prevalence of transactional sex (4.34%), whereas the highest prevalence was found from 2011 to 2015 (32.77%). This difference might be due to the publication year, the study population, or the sample size. The studies were conducted from 2000 to 2005, and the source of the data was the demography and health survey [44]. It had a large sample size, and the study population was women between the ages of 15 and 49. In contrast to other categories of the year of publication, studies from 2011 to 2015 [35, 37, 39, 42] were conducted in a single area with a relatively small sample size, and the study population was mostly young women.

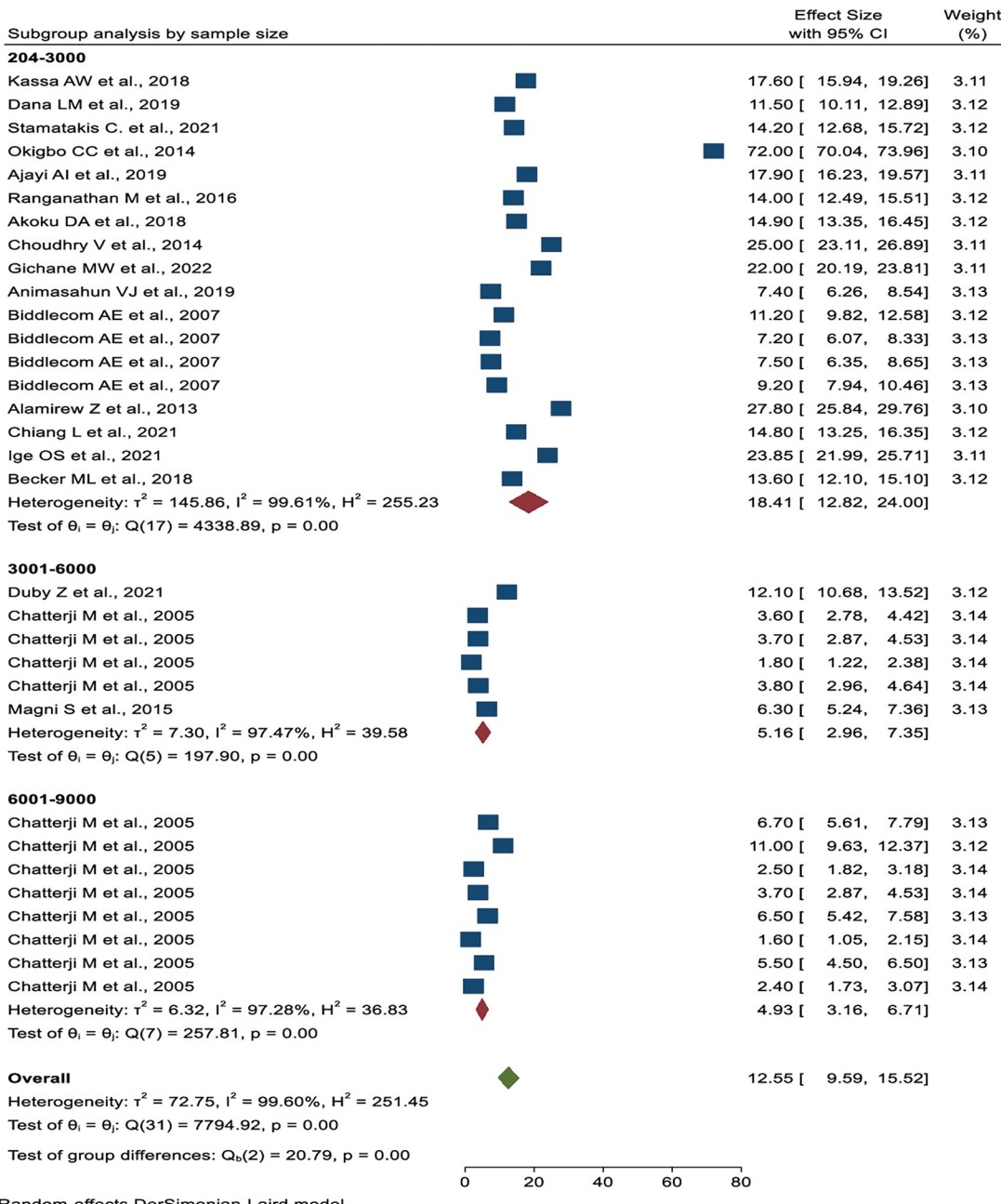

**Fig 6. Subgroup analysis of the pooled prevalence of transactional sex among women based on the sample size in Sub-Saharan Africa.**

Studies conducted with a sample size of less than 3,000 had the highest pooled prevalence of transactional sex (18.41%). In this subgroup, the study population was young and adolescent women, and they were studied in a single specific area with a small sample size. This finding was in line with the study done in China (16.5%) [58]. In contrast, a sample size greater than 6,000 had the lowest prevalence (4.93%). In terms of data source, studies conducted in a single specific area had the highest pooled prevalence of transactional sex (21.02%), while studies with large amounts of DHS data had the lowest (4.34%).

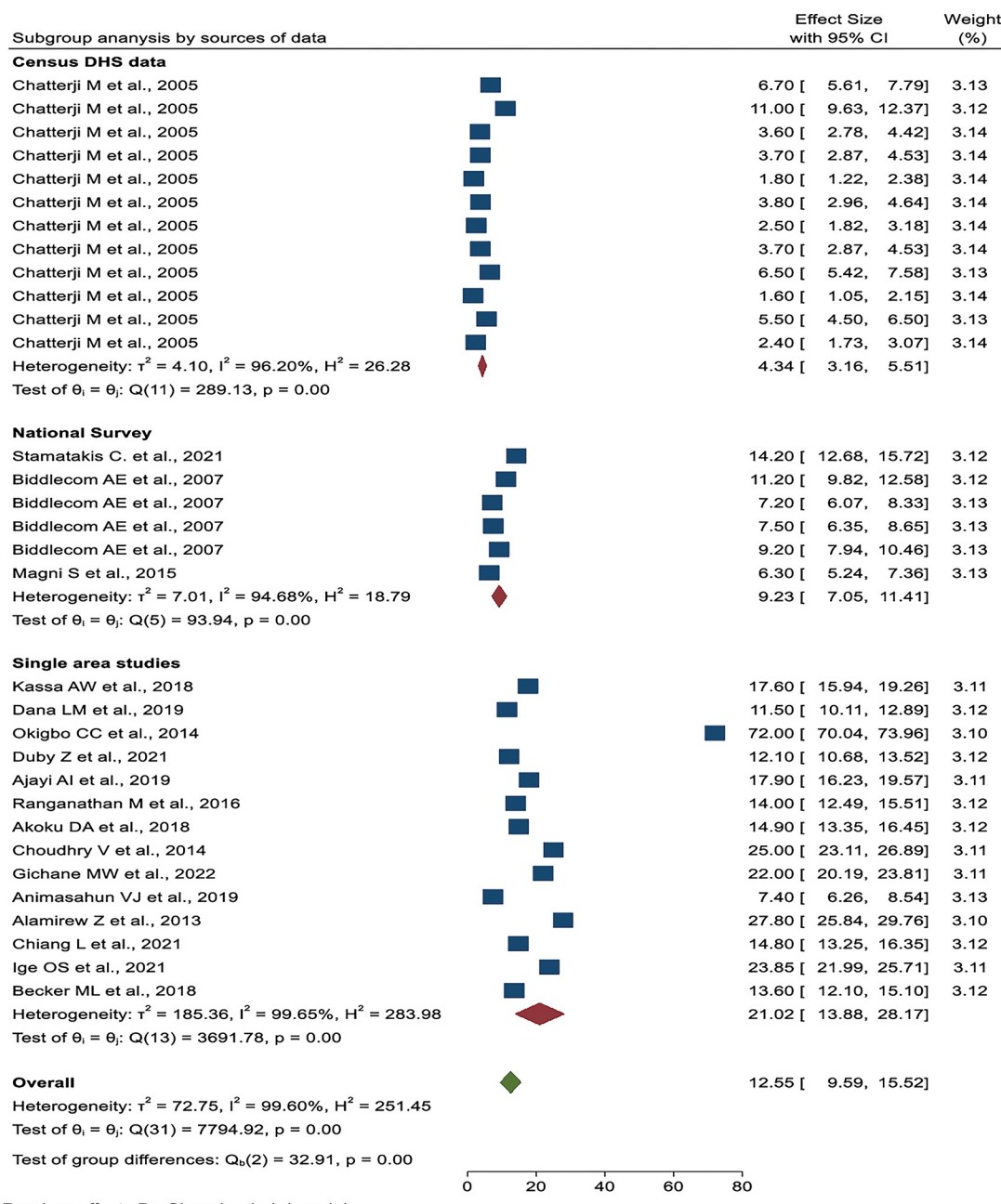

**Fig 7. Subgroup analysis of the pooled prevalence of transactional sex among women based on the source of data in Sub-Saharan Africa.**

East Africa has the greatest prevalence of transactional sex (15.57%) as compared to the other regions of sub-Saharan Africa (Central Africa, South Africa, and West Africa). Central Africa has the lowest prevalence (7.01%). This might be due to the limited number of primary studies in this region.

Among nine associated variables, eight variables were associated with the transactional sex among women in this meta-analysis. However, one variable (participants' age) was not associated with transactional sex. The odds of transactional sex were higher in participants who had

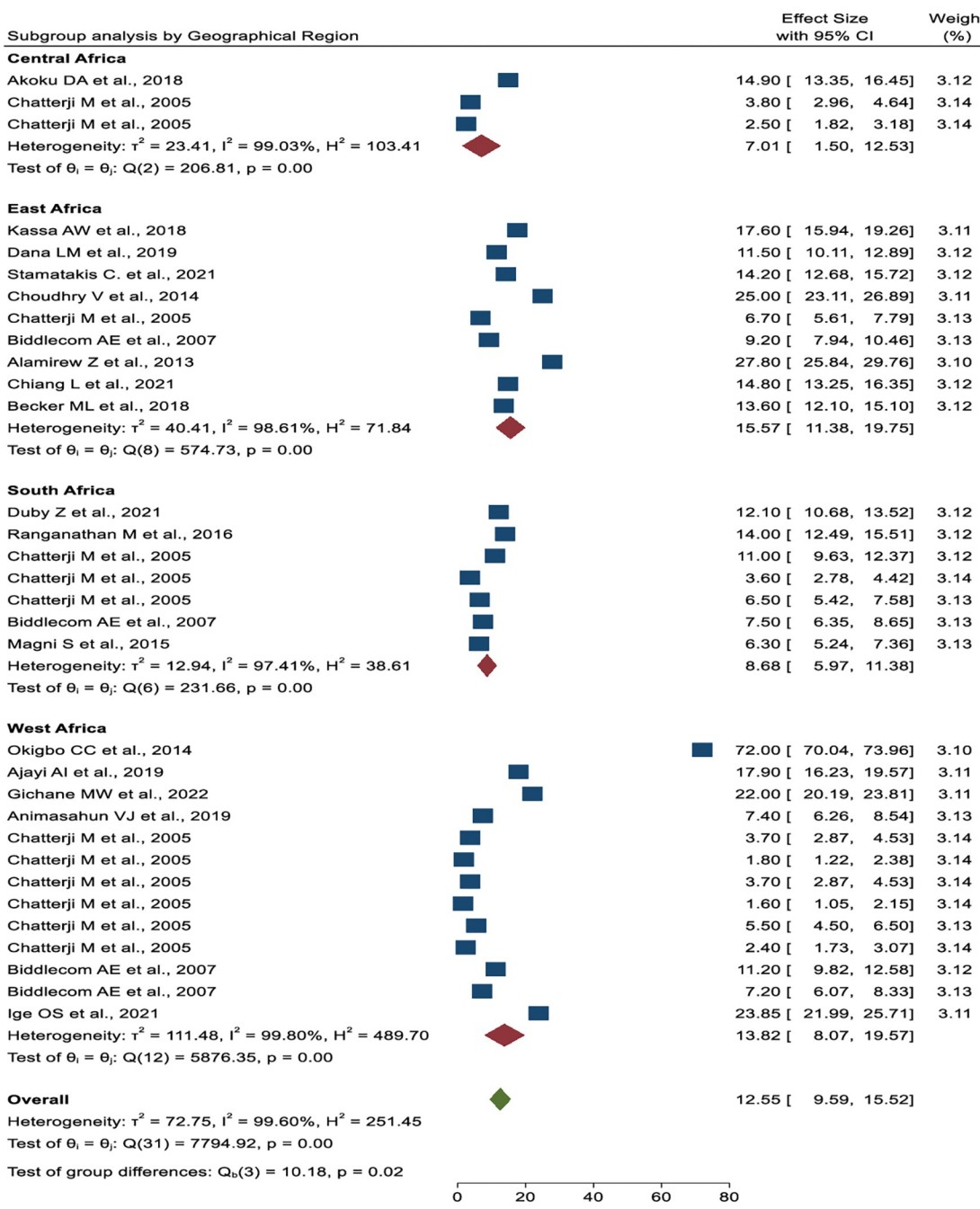

**Fig 8. Subgroup analysis of the pooled prevalence of transactional sex among women based on the geographical area in Sub-Saharan Africa.**

used alcohol as compared with participants who did not use alcohol. Some studies showed that alcohol consumision affected people's ability to feel sexual stimulation. Evidence showed that alcohol using women were more likely to engage in sexual activities, have numerous sexual partners, and engage in sex trading [59, 60]. Another study revealed that alcohol users are more likely than the general population to engage in risky sexual behaviours [61]. Other studies showed that, in females, drinking alcohol raises testosterone levels, which increase women's

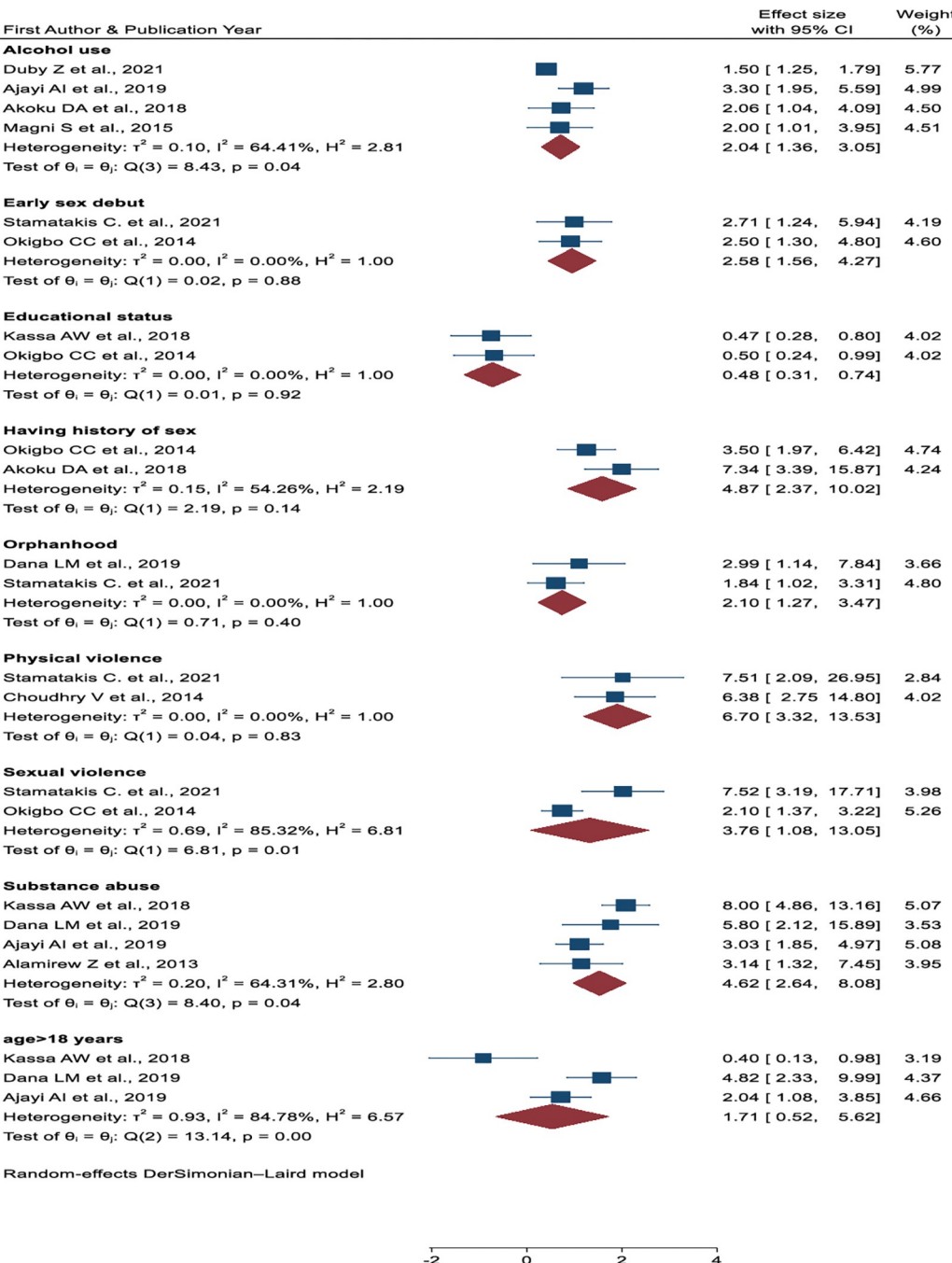

**Fig 9. Forest plot of the association between educational status, orphanhood, participants' age, alcohol use, substance abuse, having a history of sexual experience, early sex debut with, physical violence, and sexual violence with transactional sex among women in Sub-Saharan Africa.**

sexual desire [62–64]. It might be one of the reasons why women report having higher sexual desire after drinking.

Women who had an early sexual debut (first sexual initiation before the age of 16) were significantly associated with the practice of transactional sex as compared to women who had their first sexual intercourse after 16 years. Evidence shows that young women who

participated in early sexual debuts had an informal exchange of money or material items [65]. Early sexual debut has been associated with multiple sexual partners, an increased risk of unwanted pregnancy, and a higher risk of sexually transmitted illnesses. This might lead to women being exchanged for sex for money. Evidence indicated that the early sexual debut had been highly associated with selling sex [57, 66].

The educational status of the participants was found to be inversely associated with transactional sex practices. The women's educational level, having completed primary school and above made them less likely to engage in transactional sex as compared to women who had not completed primary school or had non-formal education. This could be because as women's educational levels increase, they might become more aware of the negative consequences of transactional sex.

Women who had previously engaged in sexual intercourse before giving or receiving money for sex were four times more likely to participate in transactional sex than those who had never done so. Sexual risk behaviours, especially in early adolescence, raise the chance of a variety of unfavourable health consequences as well as psychopathologies, such as increased substance use and depression.

Exposure to physical violence was associated with increased odds of practicing transactional sex. Women who had been exposed to physical violence were nearly seven times more likely to engage in exchange sex as compared to women who had no history of physical violence. As the study showed, physical violence against young women is a form of childhood trauma that is associated with negative mental and physical health outcomes, including an increased likelihood of engaging in risky sexual behaviour [67].

Sexually violent women were more likely to have transactional sex than non-violent women. Sexual violence against women is veiled in stigma and concealment, and it is fueled by damaging social norms and gender inequalities. Sexual violence includes incest, rape, and sexual violence in the context of dating or intimate relationships, sexual exploitation, internet sexual abuse, and non-contact sexual abuse. This leads to risky sexual behaviour in women and the life experiences of street women in the city. The study revealed that sexual violence has been increasing the exchange of sex [68].

Women who had substance abuse problems (Khat chew, morphine, heroin, cocaine, and shisha abusers) were four times more likely to engage in sexual activity in exchange for sex than non-abusers. Researchers discovered that using substances regularly increased the chance of sex and the number of sex partners. Women who use cocaine, prescription drugs (such as opiates and stimulants), and other illegal substances have higher sexual risk behaviours [69–72]. There is a need to escape psychological trauma, and the stresses of daily life are referred to as substance addiction. As a result, some people may start engaging in high-risk sexual behaviours including unprotected intercourse, which can lead to unintended pregnancy or sexually transmitted diseases [73].

Women who had neither one nor both parents were two times more likely to *practice* transactional sex as compared to women who had both parents. The reason is probably that the parents might serve as teachers and role models for their children, and the children would understand what is good and bad about transactional sex.

## Strengths and limitations of the study

This meta-analysis and systematic review were based on a thorough search, and studies were independently screened and extracted, which reduced the possibility of publication bias. All sections of the manuscript were written based on the PRISMA guidelines, and the quality of each study was assessed using the Newcastle-Ottawa Scale quality assessment tool. Although

we found many studies to assess the magnitude of transactional sex in Sub-Saharan Africa, we could not get studies from all countries, which might affect its representativeness. The original studies were self-reported (which might be underreported due to social desirability bias), so the pooled prevalence might be greater than this figure.

## Conclusions and recommendations

The prevalence of transactional sex among women in Sub-Saharan Africa was high. Alcohol consumption, substance abuse, early sex debuts, having a history of sexual experiences, physical violence, and sexual violence increased the practice of transactional sex. Whereas education levels greater than primary school and above reduce the practice of sex for exchange money.

## Supporting information

**S1 Checklist. PRISMA 2020 checklist.**
(DOCX)

**S1 File. A searching strategy for the prevalence of transactional sex and associated factors among women in Sub-Saharan Africa, 2022.**
(DOCX)

**S2 File. Newcastle-Ottawa Quality Assessment Scale for cross-sectional studies to assess for prevalence and associated factors of transactional sex among women in Sub-Saharan Africa, 2022.**
(DOCX)

## Acknowledgments

We would like to thank all the primary research authors and publishers.

## Author Contributions

**Conceptualization:** Gedefaye Nibret Mihretie, Alemu Degu Ayele, Agernesh Dereje Miskr, Melkalem Mamuye Azanaw, Mulugeta Dile Worke.

**Data curation:** Gedefaye Nibret Mihretie, Alemu Degu Ayele, Tewachew Muche Liyeh, Agernesh Dereje Miskr, Binyam Minuye, Melkalem Mamuye Azanaw, Mulugeta Dile Worke.

**Formal analysis:** Gedefaye Nibret Mihretie, Binyam Minuye, Melkalem Mamuye Azanaw, Mulugeta Dile Worke.

**Funding acquisition:** Gedefaye Nibret Mihretie.

**Investigation:** Gedefaye Nibret Mihretie, Alemu Degu Ayele, Tewachew Muche Liyeh, Habtamu Gebrehana Belay, Agernesh Dereje Miskr, Mulugeta Dile Worke.

**Methodology:** Gedefaye Nibret Mihretie, Alemu Degu Ayele, Agernesh Dereje Miskr, Binyam Minuye, Melkalem Mamuye Azanaw.

**Project administration:** Gedefaye Nibret Mihretie, Bekalu Getnet Kassa, Habtamu Gebrehana Belay, Agernesh Dereje Miskr, Binyam Minuye.

**Resources:** Gedefaye Nibret Mihretie, Bekalu Getnet Kassa, Alemu Degu Ayele, Tewachew Muche Liyeh, Agernesh Dereje Miskr, Binyam Minuye, Melkalem Mamuye Azanaw.

**Software:** Gedefaye Nibret Mihretie, Alemu Degu Ayele, Habtamu Gebrehana Belay, Binyam Minuye, Melkalem Mamuye Azanaw.

**Supervision:** Gedefaye Nibret Mihretie, Bekalu Getnet Kassa, Tewachew Muche Liyeh, Habtamu Gebrehana Belay, Agernesh Dereje Miskr, Melkalem Mamuye Azanaw, Mulugeta Dile Worke.

**Validation:** Gedefaye Nibret Mihretie, Mulugeta Dile Worke.

**Visualization:** Gedefaye Nibret Mihretie, Tewachew Muche Liyeh, Habtamu Gebrehana Belay.

**Writing – original draft:** Gedefaye Nibret Mihretie, Tewachew Muche Liyeh, Habtamu Gebrehana Belay, Agernesh Dereje Miskr, Mulugeta Dile Worke.

**Writing – review & editing:** Gedefaye Nibret Mihretie, Tewachew Muche Liyeh, Habtamu Gebrehana Belay, Mulugeta Dile Worke.

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
