## [Decision Letter · Decision Letter 0]

13 Jul 2022

PONE-D-22-11399The Magnitude and Associated Factors of Transactional Sex among Women in Sub-Saharan Africa: A Systematic Review and Meta-Analysis.PLOS ONE

Dear Dr. Mihretie,

Thank you for submitting your manuscript to PLOS ONE. After careful consideration, we feel that it has merit but does not fully meet PLOS ONE’s publication criteria as it currently stands. Therefore, we invite you to submit a revised version of the manuscript that addresses the points raised during the review process.

We look forward to receiving your revised manuscript.

Kind regards,

Negussie Boti Sidamo

Academic Editor

PLOS ONE

Journal Requirements:

“NO authors have competing interests”

Additional Editor Comments:

General comment

This article is poorly written. It is difficult to understand what the authors wrote on the article. To make decision it is difficult by this time. I suggest the authors to give more time and read more articles on systematic and meta-analysis.

Abstract section

1. Method: Cross-sectional studies were systematically searched from March 24, 2022, to April 6, 2022, using PubMed, Google Scholar, HINARI, Cochrane Library, and grey literature. Do you mean that this study was conducted in 10 days? Do you think that adequate time?

2. Result section: the way of writing result is don’t show statically meaning Eg. 12.55% (9.59 15.52%). Educational status (OR = .48, 95%CI, 0.27, 0.69). So it is better to report risk factors together and protective factors together. In some place you use two decimal in other place you uses one decimal after point, why?

Method section

1. Do this protocol of this review was registered on PROSPERO? If yes, add the PROSPERO ID. Also attach the proposal as supplementary file.

2. Study settings: This systematic meta-analysis was conducted in Sub-Saharan Africa Countries. What does it mean systematic meta-analysis?

3. What are Sub-Saharan Africa Countries??

4. Clearly put your Criteria for study inclusion and exclusion? The inclusion and exclusion of records should be describing a PRISMA flow diagram?

5. Your Search strategy and screening methods is poorly describe, please clearly add how Your Search strategy and screening methods?

6. You said you follow the PECO (Population, Exposure, Comparison, and Outcomes) search format, please clearly show using table.

7. Data extraction: it was poorly written, would you attach your data standardized data extraction form with your finding? What kind of checklist did the authors use for data extraction, and how are you dealing with the issue of validation? Could you perhaps include an explanation in your document?

8. Quality appraisal is poorly written would you attach your data quality appraisal, please? You said that Newcastle-Ottawa Scale quality assessment tool adapted for cross-sectional study quality assessments? Why only cross-sectional?

Reviewers' comments:

Reviewer's Responses to Questions

**Comments to the Author**

1. Is the manuscript technically sound, and do the data support the conclusions?

Reviewer #1: Partly

2. Has the statistical analysis been performed appropriately and rigorously? 

Reviewer #1: Yes

3. Have the authors made all data underlying the findings in their manuscript fully available?

Reviewer #1: Yes

4. Is the manuscript presented in an intelligible fashion and written in standard English?

Reviewer #1: No

5. Review Comments to the Author

Reviewer #1: General comments

Everyone will barely read your work because of typographical and grammatical errors, which should be corrected before proceeding.

Pooling national surveys, EDHS articles, and primary studies is a critical issue that authors should consider.

Specific comments

1. According to the authors, in addition to standard database searches, an unpublished literature (grey literature) search was conducted. The report, on the other hand, should state the type of grey literature sources used and how the search was conducted. I am afraid the grey literature on local shelves, as well as any other sources you used, does not provide enough information to fully comprehend what the investigators did.

2. What kind of articles did you find in the Cochrane Library for this review?

3. What kind of checklist did the authors use for data extraction, and how are you dealing with the issue of validation? Could you perhaps include an explanation in your document? You mentioned it was developed solely by authors? Did you utilize a guide line to help you?

4. The method section is not well developed. For example, how many authors evaluate one article in the Quality Assessment section? Is it appropriate for all authors to give a score to a single article?

5. The flow diagram you used does not follow the PRISMA guidelines. Please reconstruct it.

6. I am very concerned about the primary studies that the authors pooled in this review. Some of the studies were national surveys, such as the EDHS, while others were primary studies conducted in a single study area. Pooling these studies would introduce magnitude bias. So, how do the authors explain this significant issue?

7. What lessons can policymakers learn from your discussion? Your argument is not clearly understood. It's all about comparing and contrasting. Your argument should be based on an examination of "what treatments, tactics, or causes reduce or raise the prevalence." What successful treatments have been implemented in areas with a low magnitude? What treatments are left in areas where the magnitude is high...because we are in the twenty-first century? We place a greater focus on analytical explanations.

6. PLOS authors have the option to publish the peer review history of their article (what does this mean?). If published, this will include your full peer review and any attached files.

Reviewer #1: No

---

## [Author Response · Author response to Decision Letter 0]

9 Aug 2022

Title: Transactional Sex among Women in Sub-Saharan Africa: A Systematic Review and Meta-Analysis.

 Dear editors, editors' staffs, and reviewers, thank you once more. 

Based on your comments and queries, we tried to modify and respond as below.

 Editor Comments:

General comment

This article is poorly written. It is difficult to understand what the authors wrote on the article. To make decision it is difficult by this time. I suggest the authors to give more time and read more articles on systematic and meta-analysis.

Response: Thank you very much for your general comments. We tried to address the comments.

Abstract section

1. Method: Cross-sectional studies were systematically searched from March 24, 2022, to April 6, 2022, using PubMed, Google Scholar, HINARI, Cochrane Library, and grey literature. Do you mean that this study was conducted in 10 days? Do you think that adequate time?

Response: Thank you. Excuse us for the editorial error. It is corrected as it was searched from March 6, 2022, to April 24, 2022.

2. Result section: the way of writing result is don’t show statically meaning Eg. 12.55% (9.59 15.52%). Educational status (OR = .48, 95%CI, 0.27, 0.69). So it is better to report risk factors together and protective factors together. In some place you use two decimal in other place you uses one decimal after point, why?

 Response: Thank you, dear editorial team members. We tried to correct it.

Method section

1. Do this protocol of this review was registered on PROSPERO? If yes, add the PROSPERO ID. Also, attach the proposal as supplementary file.

Response: The International Prospective Register of Systematic Reviews has registered the review as a protocol (CRD42022323168).

2. Study settings: This systematic meta-analysis was conducted in Sub-Saharan Africa Countries. What does it mean systematic meta-analysis?

 Response: we apologies. This systematic review and meta-analysis was conducted in Sub-Saharan African countries.

3. What are Sub-Saharan Africa Countries??

 Response: it is corrected as Sub-Saharan African countries

4. Clearly, put your Criteria for study inclusion and exclusion? The inclusion and exclusion of records should be describing a PRISMA flow diagram.

Response: we tied to correct it (Figure 1).

5. Your Search strategy and screening methods is poorly describe, please clearly add how Your Search strategy and screening methods?

Response: the searching strategy was done based on Boolean operators from different databases (see the supplementary file 1). The screening method was explained in the PRISMA flow diagram (fig 1).

6. You said you follow the PECO (Population, Exposure, Comparison, and Outcomes) search format, please clearly show-using table.

 Response: Thank you. We stated the comments (PECO) as the following table.

Population Exposure Comparison Outcomes 

Women Orphanhood Women’s have no either one or both parents Women’s have both parents Practice of transactional sex 

Women Age Women age ≥18 years Women age < 18 years Practice of transactional sex 

Women Educational status women have formal education Participants’ have no formal education Practice of transactional sex 

Women Alcohol use Alcohol user Non-alcohol user Practice of transactional sex 

Women Substance abuse Substance abuser(chat chewing, cocaine, cigarette smoking , morphine, shisha) women Non users Practice of transactional sex 

Women Early sex debut Women age less than 16 years Women’s age greater than or equal to 17 years Practice of transactional sex 

Women Having history of sexual experiences Women who had sexual history before they engaged in transactional sex Women who had no sexual history before they engaged in transactional sex Practice of transactional sex 

Women Physical violence Women who had a history of physical violence before engaging in transactional sex Women who did not have a history of physical violence before engaging in transactional sex Practice of transactional sex 

Women Sexual violence Women who had a history of sexual violence before engaging in transactional sex Women who did not have a history of sexual violence before engaging in transactional sex Practice of transactional sex 

7. Data extraction: it was poorly written, would you attach your data standardized data extraction form with your finding? What kind of checklist did the authors use for data extraction, and how are you dealing with the issue of validation? Could you perhaps include an explanation in your document?

Response: Response: the whole authors do the data abstraction by the following checklists. Score differences between the investigators were managed by taking the average score of their quality evaluation outcomes. Studies that scored greater than 5/10 (>50%) based on the checklist were included in the review. 

Newcastle-Ottawa Quality Assessment Scale for cross-sectional studies to assess for prevalence and associated factors of transactional sex among women in Sub-Saharan Africa, 2022. Example quality assessment. 

 Authors Representativeness Sample size None-response rate Ascertainment Comparability Outcome Quality score

 Kassa AW et al. (2018) 1 1 1 2 2 1 8

Dana LM et al. (2019) 1 1 1 2 1 1 7

Stamatakis C. et al. (2021) 2 1 1 2 1 1 8

Okigbo CC et al. (2014) 1 1 1 1 1 1 7

Duby Z et al. (2021) 2 1 1 1 1 1 7

Ajayi AI et al. (2019) 1 2 1 1 2 1 8

Interpretation of the score

Very Good Studies: 9-10 points, Good Studies: 7-8 points, Satisfactory Studies: 5-6 points, Unsatisfactory Studies: 0 to 4 point

8. Quality appraisal is poorly written would you attach your data quality appraisal, please? You said that Newcastle-Ottawa Scale quality assessment tool adapted for cross-sectional study quality assessments. Why only cross-sectional?

Response: thank you. Because all included studies were cross-sectional studies. Example of data quality appraisal

 Authors Representativeness Sample size None-response rate Ascertainment Comparability Outcome Quality score

 Study1 1 1 1 2 2 1 8

Study2 1 1 1 2 1 1 7

Study3 2 1 1 2 1 1 8

Study4 1 1 1 1 1 1 7

Study5 2 1 1 1 1 1 7

Study6 1 2 1 1 2 1 8

Reviewer comments

Reviewer #1: 

 General comments

Everyone will barely read your work because of typographical and grammatical errors, which should be corrected before proceeding.

Response: thank dear reviewer, we tried to correct it.

Pooling national surveys, EDHS articles, and primary studies is a critical issue that authors should consider.

Response: Although these are different studies, we tried to do a subgroup analysis by the source of the data to identify the potential risk of bias.

 Specific comments

1. According to the authors, in addition to standard database searches, an unpublished literature (grey literature) search was conducted. The report, on the other hand, should state the type of grey literature sources used and how the search was conducted. I am afraid the grey literature on local shelves, as well as any other sources you used, does not provide enough information to fully comprehend what the investigators did.

Response: Thank you, dear Reviewer. The grey literature was considered because the titles were similar to the outcome of interest. However, due to insufficient information, it was rejected during screening.

2. What kind of articles did you find in the Cochrane Library for this review?

Response: Thank you. We tried searching the Cochrane library, but we did not find inclusive data because it mainly contains randomized control trial studies. The primary studies included in this systematic review and meta-analysis was mainly self-reported proportions, which might not be found in Cochraine Liberary.

3. What kind of checklist did the authors use for data extraction, and how are you dealing with the issue of validation? Could you perhaps include an explanation in your document? You mentioned it was developed solely by authors? Did you utilize a guideline to help you?

Response: the whole authors do the data abstraction by the following checklists. Score differences between the investigators were managed by taking the average score of their quality evaluation outcomes. Studies that scored greater than 5/10 (>50%) based on the checklist were included in the review. 

Newcastle-Ottawa Quality Assessment Scale for cross-sectional studies to assess for prevalence and associated factors of transactional sex among women in Sub-Saharan Africa, 2022. Example 

 Authors Representativeness Sample size None-response rate Ascertainment Comparability Outcome Quality score

 Kassa AW et al. (2018) 1 1 1 2 2 1 8

Dana LM et al. (2019) 1 1 1 2 1 1 7

Stamatakis C. et al. (2021) 2 1 1 2 1 1 8

Okigbo CC et al. (2014) 1 1 1 1 1 1 7

Duby Z et al. (2021) 2 1 1 1 1 1 7

Ajayi AI et al. (2019) 1 2 1 1 2 1 8

Interpretation of the score

Very Good Studies: 9-10 points, Good Studies: 7-8 points, Satisfactory Studies: 5-6 points, Unsatisfactory Studies: 0 to 4 point

4. The method section is not well developed. For example, how many authors evaluate one article in the Quality Assessment section? Is it appropriate for all authors to give a score to a single article?

Response: all authors evaluated each articles. This reduces selection bias.

5. The flow diagram you used does not follow the PRISMA guidelines. Please reconstruct it.

Response: corrected

6. I am very concerned about the primary studies that the authors pooled in this review. Some of the studies were national surveys, such as the EDHS, while others were primary studies conducted in a single study area. Pooling these studies would introduce magnitude bias. So, how do the authors explain this significant issue?

Response: ok thank you. There are differences b/n national survey, DHS data and single area studies. There may be magnitude bias. However, to show magnitude bias effect, we tried to do sub group analysis based on the sources of data (census DHS data, National survey and and small area primary studies). Small area primary studies have highest magnitude. (See Fig 5c)

7. What lessons can policymakers learn from your discussion? Your argument is not clearly understood. It is all about comparing and contrasting. Your argument should be based on an examination of "what treatments, tactics, or causes reduce or raise the prevalence." What successful treatments have been implemented in areas with a low magnitude? What treatments are left in areas where the magnitude is high...because we are in the twenty-first century? We place a greater focus on analytical explanations.

Response: thank you, based the finding, intervention should be implemented. Governmental and other stakeholders are designed to reduce alcohol utilization, provide health information about the negative consequences of early sex debut, substance abuse, and reduce sexual violence, ensuring gender equality through mass media, which should be included in state policy. Furthermore, we recommended mixed qualitative and quantitative studies, to answer why questions. 

 Thank You

---

## [Decision Letter · Decision Letter 1]

3 Feb 2023

PONE-D-22-11399R1Transactional Sex among Women in Sub-Saharan Africa: A Systematic Review and Meta-Analysis.PLOS ONE

Dear Dr. Mihretie,

Thank you for submitting your manuscript to PLOS ONE. After careful consideration, we feel that it has merit but does not fully meet PLOS ONE’s publication criteria as it currently stands. Therefore, we invite you to submit a revised version of the manuscript that addresses the points raised during the review process.

We look forward to receiving your revised manuscript.

Kind regards,

Felix Bongomin, MB ChB, MSc, MMed, FECMM

Academic Editor

PLOS ONE

Reviewers' comments:

Reviewer's Responses to Questions

**Comments to the Author**

1. If the authors have adequately addressed your comments raised in a previous round of review and you feel that this manuscript is now acceptable for publication, you may indicate that here to bypass the “Comments to the Author” section, enter your conflict of interest statement in the “Confidential to Editor” section, and submit your "Accept" recommendation.

Reviewer #2: (No Response)

2. Is the manuscript technically sound, and do the data support the conclusions?

Reviewer #2: Partly

3. Has the statistical analysis been performed appropriately and rigorously? 

Reviewer #2: Yes

4. Have the authors made all data underlying the findings in their manuscript fully available?

Reviewer #2: Yes

5. Is the manuscript presented in an intelligible fashion and written in standard English?

Reviewer #2: Yes

6. Review Comments to the Author

Reviewer #2: I appreciate for the authors addressed important points for this public health concern. The paper needs to improve the Editorial errors, spelling, and scientific writing. This paper failed to address information on transactional sex from the globe to the SSA in the introduction section. The last paragraph of the introduction is too long and not targeted, focusing on the gap and the aim of the study (avoid the significance of the study); the method section is good but luck some clarity and is not written scientifically for example exclusion criteria, population; operational definition; avoid using unscientific words, for example, irrelevant target population; result section paragraph one-line 4; one study from each country (Zambia, Zimbabwe, Benin, Burkina Faso, Central Africa Republic(CAR), Chad, Guinea, Niger, and Togo)[36], you used many studies but you cited only one reference. Discussion paragraphs one and two are out of your context (unnecessary in the discussion part); Studies conducted from 2000 to 2005, observed the lowest pooled prevalence of transactional sex (4.34%) in subgroup analysis….this part has no citation of references. The discussion part need to be written about the maginitude and factors by comparing your finding with others and justifying possible resons and also you should cite the referances. The conclusion is general and failed to concluded based on the pertinent finding of the study and you should also wriwrite the recommendation.

7. PLOS authors have the option to publish the peer review history of their article (what does this mean?). If published, this will include your full peer review and any attached files.

Reviewer #2: No

---

## [Author Response · Author response to Decision Letter 1]

7 Feb 2023

Title: Transactional Sex among Women in Sub-Saharan Africa: A Systematic Review and Meta-Analysis.

 Dear editor and reviewers, thank you once more. 

We tried to amend and answer as follows in response to your comments and questions.

 Editor Comments:

Abstract section

1. Method: Cross-sectional studies were systematically searched from March 24, 2022, to April 6, 2022, using PubMed, Google Scholar, HINARI, Cochrane Library, and grey literature. Do you mean that this study was conducted in 10 days? Do you think that adequate time?

Response: Thank you. Excuse us for the editorial error. It is corrected as it was searched from March 6, 2022, to April 24, 2022.

2. Result section: the way of writing result is don’t show statically meaning Eg. 12.55% (9.59 15.52%). Educational status (OR = .48, 95%CI, 0.27, 0.69). So it is better to report risk factors together and protective factors together. In some place you use two decimal in other place you uses one decimal after point, why?

 Response: Thank you, dear editorial team members. We tried to correct it.

Method section

1. Do this protocol of this review was registered on PROSPERO? If yes, add the PROSPERO ID. Also, attach the proposal as supplementary file.

Response: The International Prospective Register of Systematic Reviews has registered the review as a protocol (CRD42022323168).

2. Study settings: This systematic meta-analysis was conducted in Sub-Saharan Africa Countries. What does it mean systematic meta-analysis?

 Response: we apologies. This systematic review and meta-analysis was conducted in Sub-Saharan African countries.

3. What are Sub-Saharan African Countries?

 Response: Sub-Saharan Africa is those countries of the African continent that are not considered part of North Africa. Sometimes referred to as area of Black Africa. List of Sub-Saharan African Countries includes Angola, Benin, Botswana, Burkina Faso, Burundi, Cameroon Cape Verde, Chad, Central African Republic, Comoros, Congo , Côte d'Ivoire, Djibouti, Equatorial Guinea, Eritrea, Ethiopia, Gabon, Gambia, Ghana, Guinea, Guinea-Bissau, Kenya, Lesotho, Liberia, Madagascar, Malawi, Mali, Mauritania, Mozambique, Namibia, Niger, Nigeria, Rwanda, Senegal, Seychelles, Sierra Leone, Somalia, South Africa, Sudan, Tanzania, Togo, Uganda, Western Sahara, Zambia,Zimbabwe. 

4. Clearly, put your Criteria for study inclusion and exclusion? The inclusion and exclusion of records should be describing a PRISMA flow diagram.

Response: we included in PRIAMA flow diagram (Figure 1).

5. Your Search strategy and screening methods is poorly describe, please clearly add how Your Search strategy and screening methods?

Response: thank you very much for the comments. We tried to correct he comments. (See the method section of the main manuscript)

6. You said you follow the PECO (Population, Exposure, Comparison, and Outcomes) search format, please clearly show-using table.

 Response: Thank you. We stated the comments (PECO) as the following table.

Population Exposure Comparison Outcomes 

Women Orphanhood Women’s have no either one or both parents Women’s have both parents Practice of transactional sex 

Women Age Women age ≥18 years Women age < 18 years Practice of transactional sex 

Women Educational status women have formal education Participants’ have no formal education Practice of transactional sex 

Women Alcohol use Alcohol user Non-alcohol user Practice of transactional sex 

Women Substance abuse Substance abuser(chat chewing, cocaine, cigarette smoking , morphine, shisha) women Non users Practice of transactional sex 

Women Early sex debut Women age less than 16 years Women’s age greater than or equal to 17 years Practice of transactional sex 

Women Having history of sexual experiences Women who had sexual history before they engaged in transactional sex Women who had no sexual history before they engaged in transactional sex Practice of transactional sex 

Women Physical violence Women who had a history of physical violence before engaging in transactional sex Women who did not have a history of physical violence before engaging in transactional sex Practice of transactional sex 

Women Sexual violence Women who had a history of sexual violence before engaging in transactional sex Women who did not have a history of sexual violence before engaging in transactional sex Practice of transactional sex 

7. Data extraction: it was poorly written, would you attach your data standardized data extraction form with your finding? What kind of checklist did the authors use for data extraction, and how are you dealing with the issue of validation? Could you perhaps include an explanation in your document?

Response: Thank you the comments. The full texts of the studies were screened based on objectives, methods, participants, and the outcomes. We revised it based on the comments (see page 6&7 of the manuscript). 

8. Quality appraisal is poorly written would you attach your data quality appraisal, please? You said that Newcastle-Ottawa Scale quality assessment tool adapted for cross-sectional study quality assessments. Why only cross-sectional?

Response: thank you. We attached Newcastle-Ottawa Scale quality assessment tool as supplementary file2 (see supplementary file2). During searching, we found only cross-sectional studies.

General comment from the Editor

This article is poorly written. It is difficult to understand what the authors wrote on the article. To make decision it is difficult by this time. I suggest the authors to give more time and read more articles on systematic and meta-analysis 

Response: Thank you very much for your general comments. We tried to revise the comments.

Reviewer Comments to the Author

Reviewer #2: I appreciate for the authors addressed important points for this public health concern.

Response: Thank very much

1. The paper needs to improve the Editorial errors, spelling, and scientific writing. 

Response: thank you very much. We tried to do so and amended it based on the comments.

2. Introduction: This paper failed to address information on transactional sex from the globe to the SSA in the introduction section. The last paragraph of the introduction is too long and not targeted, focusing on the gap and the aim of the study (avoid the significance of the study).

Response: Thank you the comments. We tried to amend it (see the introduction of the manuscript)

3. Method section is good but luck some clarity and is not written scientifically. For example exclusion criteria, population; operational definition; avoid using unscientific words, for example, irrelevant target population.

Response: Thank you dear reviewer. We corrected it (see the manuscript).

4. Results section paragraph one-line 4; one study from each country (Zambia, Zimbabwe, Benin, Burkina Faso, Central Africa Republic (CAR), Chad, Guinea, Niger, and Togo) [36], you used many studies but you cited only one reference.

Response: Thank you your comments. The same authors conducted these many studies at the same time (see Ref. 45).

5. Discussion paragraphs one and two are out of your context (unnecessary in the discussion part); Studies conducted from 2000 to 2005, observed the lowest pooled prevalence of transactional sex (4.34%) in subgroup analysis….this part has no citation of references. The discussion part need to be written about the magnitudes and factors by comparing your finding with others and justifying possible reasons and you should cite the references. 

Response: Thank you for interesting comments. We cited the references and corrected it. 

6. The conclusion is general and failed to concluded based on the pertinent finding of the study and you should also rewrite the recommendation

Response: Thank you for your feedback. Since the studies were conducted in sub-Saharan African countries, the recommendations are for each of the countries and for sub-Saharan Africa as a whole.

 Thank you very much with regards

---

## [Decision Letter · Decision Letter 2]

17 Apr 2023

PONE-D-22-11399R2Transactional Sex among Women in Sub-Saharan Africa: A Systematic Review and Meta-Analysis.PLOS ONE

Dear Dr. Mihretie,

Thank you for submitting your manuscript to PLOS ONE. After careful consideration, we feel that it has merit but does not fully meet PLOS ONE’s publication criteria as it currently stands. Therefore, we invite you to submit a revised version of the manuscript that addresses the points raised during the review process.

We look forward to receiving your revised manuscript.

Kind regards,

Felix Bongomin, MB ChB, MSc, MMed, FECMM

Academic Editor

PLOS ONE

Journal Requirements:

Reviewers' comments:

Reviewer's Responses to Questions

**Comments to the Author**

1. If the authors have adequately addressed your comments raised in a previous round of review and you feel that this manuscript is now acceptable for publication, you may indicate that here to bypass the “Comments to the Author” section, enter your conflict of interest statement in the “Confidential to Editor” section, and submit your "Accept" recommendation.

Reviewer #3: (No Response)

2. Is the manuscript technically sound, and do the data support the conclusions?

Reviewer #3: Yes

3. Has the statistical analysis been performed appropriately and rigorously? 

Reviewer #3: Yes

4. Have the authors made all data underlying the findings in their manuscript fully available?

Reviewer #3: Yes

5. Is the manuscript presented in an intelligible fashion and written in standard English?

Reviewer #3: No

6. Review Comments to the Author

Reviewer #3: I thank the authors for this interesting manuscript. However, I have a few comments.

1. The manuscript still has grammatical errors. Please read through the entire paper carefully to correct these.

2. The introduction section should be revised to flow better from i) overview of the subjects "Transactional sex" to ii) global statistics and statistics concerning sub-Saharan Africa, iii) what is already known and iv) highlight the current existing gaps in this context.

The section has lots of mix-ups of tenses. please correct that as well.

Reference these statements; "A significant proportion of females had multiple concurrent sexual relationships, including "sugar daddies," and engaged in risky sex. Because of their risky sexual practices, the girls and their

sexual partners, including schoolmates, were at risk of HIV infection and other sexually

transmitted infections (STIs)"

Also, rephrase the statements to harmonise the transition from the previous paragraph to the next paragraph.

3. Discussion section; The statement "The finding of this study was inlined with the study shown in high-income countries (10%)" is not clear. Perhaps you could state,"Our findings are comparable to findings from a study by XXX and colleagues where the prevalence of transactional sex in high income countries was estimated at 10%......"

Also, summarize the following statements into about 2 statements with the relevant information. " In Cameroon, 30% of girls (aged

15–20) had ever engaged in sexual relations in exchange for money or gifts[50], whereas in Malawi

it was approximately 66% of girls aged 10–18[51]. In urban Tanzania, 80% of girls (aged 14–19)

answered positively to a question about ever receiving money from boyfriends for sex[52].

In a study of university women (aged 16 years and older) in Nigeria, 18% reported ever having

exchanged sex for money, gifts, or favors[51]. A study in Kenya found that 78 percent of girls

(aged 15-19) usually have transactional sex[53]. Seven percent of youth in Canada, reported

having bought and sold such services for sex in their lifetime[54]. For adolescent participants in

Sweden, 1.5% of the girls showed that they had sold sex for money or other repayments[55]. In

America, the prevalence of transactional sex was 57 %[56], and in Norway exchanged sex was

1.4% among adolescents[57]. This finding is also similar to the study conducted on African

American women (13.1%)[58]."

7. PLOS authors have the option to publish the peer review history of their article (what does this mean?). If published, this will include your full peer review and any attached files.

Reviewer #3: **Yes: **Winnie Kibone

---

## [Author Response · Author response to Decision Letter 2]

24 May 2023

Title: Transactional Sex among Women in Sub-Saharan Africa: A Systematic Review and Meta-Analysis.

 Dear editor and reviewers, thank you once more. 

We tried to amend the manuscript based on the given comments as well as the whole manuscript.

Review Comments to the Author

Reviewer #3: I thank the authors for this interesting manuscript. However, I have a few comments.

1. The manuscript still has grammatical errors. Please read the entire paper carefully to correct these.

Response: thank you, we tried to review the whole manuscript and amend it

2. The introduction section should be revised to flow better from i) overview of the subjects "Transactional sex" to ii) global statistics and statistics concerning sub-Saharan Africa, iii) what is already known and iv) highlight the current existing gaps in this context.The section has many mix-ups of tenses. Please correct that as well.

Response: we revised and amend it 

3. Reference these statements; "A significant proportion of females had multiple concurrent sexual relationships, including "sugar daddies," and engaged in risky sex. Because of their risky sexual practices, the girls and their sexual partners, including schoolmates, were at risk of HIV infection and other sexually transmitted infections (STIs)" Also, rephrase the statements to harmonise the transition from the previous paragraph to the next paragraph.

Response: we cited the reference 

4. Discussion section; The statement "The finding of this study was inlined with the study shown in high-income countries (10%)" is not clear. Perhaps you could state,"Our findings are comparable to findings from a study by XXX and colleagues where the prevalence of transactional sex in high income countries was estimated at 10%......"

Response: thank you very much for your input. We corrected it.

5. Also, summarize the following statements into about 2 statements with the relevant information. " In Cameroon, 30% of girls (aged 15–20) had ever engaged in sexual relations in exchange for money or gifts[50], whereas in Malawi it was approximately 66% of girls aged 10–18[51]. In urban Tanzania, 80% of girls (aged 14–19) answered positively to a question about ever receiving money from boyfriends for sex [52]. In a study of university women (aged 16 years and older) in Nigeria, 18% reported ever having exchanged sex for money, gifts, or favors [51]. A study in Kenya found that 78 percent of girls (aged 15-19) usually have transactional sex [53]. Seven percent of youth in Canada, reported having bought and sold such services for sex in their lifetime [54]. For adolescent participants in Sweden, 1.5% of the girls showed that they had sold sex for money or other repayments [55]. In America, the prevalence of transactional sex was 57 % [56], and in Norway, exchanged sex was 1.4% among adolescents [57]. This finding is also similar to the study conducted on African American women (13.1%) [58].

Response: thank valuable comments. We corrected it. (Women in Nigeria, 18% [51]; in Kenya, 78% [53]; in Canada, 7% [54]; In Sweden, 1.5% [55]; in America, 57 % [56], and in Norway, 1.4% [57] have ever exchanged sex for money, gifts, or favours).

---

## [Editor Report · Decision Letter 3]

25 May 2023

Transactional Sex among Women in Sub-Saharan Africa: A Systematic Review and Meta-Analysis.

PONE-D-22-11399R3

Dear Dr. Mihretie,

We’re pleased to inform you that your manuscript has been judged scientifically suitable for publication and will be formally accepted for publication once it meets all outstanding technical requirements.

Kind regards,

Felix Bongomin, MB ChB, MSc, MMed, FECMM

Academic Editor

PLOS ONE
---

## [Editor Report · Acceptance letter]

31 May 2023

PONE-D-22-11399R3 

Transactional Sex among Women in Sub-Saharan Africa: A Systematic Review and Meta-Analysis 

Dear Dr. Mihretie:

I'm pleased to inform you that your manuscript has been deemed suitable for publication in PLOS ONE. Congratulations! Your manuscript is now with our production department. 

Kind regards, 

on behalf of

Dr. Felix Bongomin 

Academic Editor

PLOS ONE